# Climate Adaptation Strategies for Maintaining Rice Grain Quality in Temperate Regions

**DOI:** 10.3390/biology14070801

**Published:** 2025-07-02

**Authors:** Yvonne Fernando, Ben Ovenden, Nese Sreenivasulu, Vito Butardo

**Affiliations:** 1Department of Chemistry and Biotechnology, Swinburne University of Technology, Hawthorn, VIC 3122, Australia; ylfernando@swin.edu.au; 2NSW Department of Primary Industries, Wagga Wagga, NSW 2650, Australia; ben.ovenden@dpird.nsw.gov.au; 3International Rice Research Institute, Los Baños 4031, Philippines; n.sreenivasulu@irri.org

**Keywords:** climate adaptation, grain quality parameters, environmental stress response, *japonica* rice, phenotyping technologies

## Abstract

Climate change presents significant challenges for temperate rice production systems, affecting both grain quality and consumer markets. This review examines how environmental factors including temperature extremes, altered rainfall patterns, elevated CO_2_, and increased salinity impact key quality parameters in temperate-grown rice, with particular focus on the Australian industry in comparison to other temperate regions. The Australian rice industry has developed adaptive capacity through breeding cold-tolerant varieties, implementing precision agriculture technologies, and optimising water-efficient practices. However, projected increases in temperature variability and altered precipitation patterns will continue to challenge quality maintenance. Emerging phenotyping technologies, including hyperspectral imaging and machine learning approaches, offer promising tools for monitoring climate impacts and accelerating adaptation. This review highlights the urgent need for integrated adaptation strategies combining improved understanding of physiological responses with practical breeding and management approaches to sustain temperate rice quality under increasing environmental variability.

## 1. Introduction

Climate change fundamentally alters rice grain quality through interconnected physiological and biochemical mechanisms that operate throughout grain development. These abiotic stresses affect source–sink relationships at first, inhibiting photosynthesis in the vegetative canopy, leading to limited biomass accumulation, and in addition limit sucrose transport into developing grains and thus restricting grain development and maturation [1]. Rising temperatures modify enzyme kinetics in starch biosynthesis pathways, altering amylose to amylopectin ratios, gelatinisation properties, and protein accumulation patterns. As a result, percent protein content increases and the percent of amylose decreases under global warming conditions [2]. Concurrently, drought stress disrupts assimilate translocation and grain filling processes, while elevated CO_2_ alters carbon–nitrogen balance in developing grains. These biochemical modifications manifest as quantifiable differences in starch granules with irregular shapes and altered protein bodies, resulting in the increase in air spaces within the endosperm triggering chalk. These global changes in temperature also affect head rice yield due to reduced grain filling duration and increase in chalky grain with partial grain filling. These biochemical alterations disturb milling quality, appearance attributes, cooking behaviour, and nutritional composition; parameters that directly determine market value and consumer acceptance [2,3,4].

Temperate rice refers to *japonica*-type rice varieties that are specifically adapted to grow in cooler climates with distinct seasonal temperature variations. These varieties are typically cultivated in regions such as East Asia, Europe, Australia, and North America [5,6]. Temperate rice cultivation, primarily of *japonica* varieties, presents distinct research challenges compared to tropical rice production systems. Characterised by cooler climates, shorter growing seasons, longer day lengths, and specific environmental vulnerabilities, temperate regions face unique climate change impacts [6,7,8]. Due to recent climate change, the night temperature has increased rapidly in comparison to high day temperatures, and thus high night temperatures (HNTs) are impacting the temperate countries too [9]. These temperate regions increasingly experience extreme temperature variability, ranging between cold stress at early plant establishment to high day and HNT during reproductive stages and altered precipitation regimes resulting in water scarcity, all directly influencing grain development and quality [10].

The Australian rice industry, primarily located in the southern Murray–Darling Basin (approximately 34° S to 36° S latitude and and 144° E to 147° E longitude) in New South Wales, exemplifies the complexities of temperate rice production under changing climatic conditions. Despite fluctuations in production influenced by water availability and competing land use pressures, this industry has developed significant adaptive capacity through genetic improvement of varieties and optimised agronomic practices [11]. However, maintaining good grain quality under increasingly variable climate conditions presents ongoing challenges requiring systematic evaluation of adaptation pathways.

Climate change creates a fundamental challenge for temperate rice production systems: maintaining grain quality characteristics valued by consumers and markets while adapting to increasingly variable environmental conditions. This review demonstrates that climate factors affect grain quality through specific biochemical pathways that modify starch structure, protein accumulation, and aroma compound synthesis during grain development. These modifications manifest differently across rice quality classes, with medium-grain *japonica* varieties showing vulnerability to heat-induced amylose reduction, aromatic varieties experiencing modified fragrance compound synthesis under drought, and long-grain types demonstrating compromised kernel integrity under combined stressors [12,13].

The adaptation pathways identified in this review, spanning genetic, agronomic, technological, and grain quality assessment aspects, demonstrate that maintaining quality under climate change requires integrated approaches rather than isolated interventions (Figure 1). Breeding programmes must target quality-specific biochemical pathways showing climate sensitivity, while water management innovations require calibration to variety-specific responses. Technological systems enhance adaptation effectiveness through improved monitoring and precision management, whilst advanced assessment methodologies accelerate the adaptation feedback cycle by providing earlier and more comprehensive quality evaluation.

Effective climate adaptation extends beyond technical approaches to encompass policy frameworks and consumer engagement strategies. Regulatory mechanisms governing water allocation, economic incentives rewarding quality maintenance, and research funding priorities all significantly influence adaptation capacity. Current policy structures often emphasise yield and production volume over grain quality and nutrition, creating potential misalignment with market positioning in premium segments. International cooperation mechanisms facilitating knowledge exchange and germplasm sharing represent underutilised policy approaches for enhancing adaptation efficiency across temperate production regions [14,15].

This review addresses three critical knowledge gaps in understanding climate change impacts on temperate rice grain quality: (1) How do temperature extremes, altered precipitation patterns, and elevated atmospheric CO_2_ specifically affect grain quality parameters in temperate *japonica* varieties compared to tropical *indica* varieties? (2) What differences exist in climate vulnerability across quality classes (medium-grain, short-grain, aromatic varieties, etc) within temperate production systems? (3) Which adaptation strategies demonstrate evidence-based effectiveness for maintaining grain quality under projected climate scenarios in temperate regions? By integrating analysis of mechanistic relationships between environmental factors and grain quality parameters, differential vulnerability across rice quality classes, and adaptation strategy effectiveness, this review provides direction for research priorities to secure high-quality rice production in temperate production zones. This review is particularly timely given the increasing global demand for high-quality rice and the pressing need for climate-resilient agricultural systems.

This review employed a structured literature search using Web of Science, Scopus, and Google Scholar, focusing on studies from 1996 to 2025 that examined climate impacts on temperate rice grain quality. Priority was given to peer-reviewed field studies in temperate regions, especially those assessing Australian rice varieties and using standardised quality assessment methods. Meta-analyses and controlled-environment studies were included, where relevant, to support mechanistic insights and fill data gaps.

## 2. Overview of the Australian Rice Industry

The Australian rice industry has evolved through distinct developmental phases that have progressively enhanced its adaptive capacity to environmental constraints, creating institutional structures, knowledge systems, and technological approaches now central to its climate change response capabilities. This historical evolution provides critical context for understanding contemporary adaptation challenges and opportunities.

### 2.1. Historical Development and Adaptive Evolution

The industry’s formative period (1850s–1920s) established fundamental geographical and genetic foundations while addressing the challenges of temperate rice cultivation in the Australian environment. Initial cultivation attempts by Chinese prospectors in the 1850s proved not commercially viable due to inadequate understanding of local agroclimatic conditions. The first documented successful cultivation occurred in Northern Queensland during the 1860s, though even these operations faced constraints from soil toxicities and pest pressures [16,17,18]. This early period demonstrated the significant climate and environmental barriers requiring adaptation for sustainable production.

The industry’s infrastructural foundation emerged with the establishment of the Murrumbidgee Irrigation Area in the early 20th century. Government policy and infrastructure development, particularly the construction of the Burrinjuck Dam, created the hydrological conditions necessary for reliable temperate rice production [17,18,19]. This confluence of infrastructure development and genetic adaptation established the geographical concentration and varietal orientation that continues to cater the need of the Australian rice industry.

The industry’s institutional foundation developed with the formation of Ricegrowers’ Co-operative Mills Limited in 1955. This cooperative structure, created in response to concerns regarding price transmission to producers, established frameworks for coordinated industry response to production challenges. The launch of the ‘Sunwhite’ rice brand emphasised quality differentiation as a core strategic approach, establishing the quality focus that remains central to industry identity and market positioning [17,18,20].

The modern adaptation phase (post-1970s) has featured increasing engagement with environmental constraints, particularly water limitations. The development of varieties with shorter growing cycles (e.g., ‘Quest’) and enhanced cold tolerance (e.g., ‘Sherpa’) exemplifies the industry’s genetic adaptation to resource constraints and temperature vulnerabilities. This period has also seen significant technological adoption, establishing Australia’s rice production systems as the world’s most water-efficient practices [17,21].

This historical development trajectory has created three critical adaptive capacity elements that influence contemporary climate response: (1) institutional knowledge systems that facilitate information transfer and coordinated adaptation; (2) genetic adaptation capacity through established breeding programmes targeting environmental constraints; and (3) technological innovation systems that systematically address resource limitations. These adaptive capacity elements provided the foundation for contemporary climate adaptation strategies while shaping the industry’s approach to emerging challenges.

The New South Wales Department of Primary Industries (NSW DPI), established in 1928, oversees agriculture, while the Yanco Agricultural Institute (YAI), in collaboration with NSW DPI, leads rice research in the Murray and Murrumbidgee Irrigation Areas. The YAI’s rice improvement program, which began in 1959, received additional investment and support in 1979, enabling further progress in varietal improvement and grain quality [19]. For decades, the NSW DPI led Australian rice breeding and established the benchmark for grain quality assessment through its dedicated facilities and protocols at the YAI. In 2022, driven by strategic goals for improved water productivity, accelerated genetic gain, and enhanced commercial focus, these responsibilities transitioned to Rice Breeding Australia (RBA), a new entity formed by AgriFutures Australia, SunRice, and the Ricegrowers’ Association of Australia (RGA). This transition was facilitated by the AgriFutures Rice Advisory Panel where the corresponding author was a member of. This shift signifies a move away from the YAI as the custodian of quality standards, with RBA adopting faster assessment methods and technologies at critical points in its accelerated breeding cycle and prioritising quality traits based on direct commercial imperatives [22].

### 2.2. Contemporary Production Systems and Market Position

The modern Australian rice industry occupies a distinctive market niche characterised by high-quality temperate *japonica* production within a highly variable production environment. Production exhibits substantial interannual fluctuations primarily driven by water availability, exemplified by the contrast between 2019–2020 (approximately 57,000 tonnes) and 2020–2021 (423,000 tonnes) despite yield reductions from cooler temperatures [23]. This production volatility creates significant challenges for industry planning and climate adaptation investments.

The industry operates within a complex agricultural landscape where crop rotation practices (integrating rice with wheat, barley, canola, or livestock) provide agronomic benefits while introducing economic complexities that influence adaptation decisions. Increasing competition from alternative crops, particularly cotton and high-value perennial tree crops such as almonds and walnuts, further complicates rice adaptation planning by reshaping economic incentives and influencing decisions on water allocation, land use, and irrigation infrastructure investment [11].

Water policy interventions create additional complexity for climate adaptation efforts. Recovery policies in the Murray–Darling Basin, implemented to address broader environmental sustainability concerns, have constrained water availability for rice cultivation, particularly during drier periods [24]. This creates a policy environment where climate adaptation efforts must address both direct climate impacts and policy-mediated effects on resource availability.

Despite production volatility, domestic rice consumption has demonstrated consistent growth over the past three decades, driven substantially by immigration from rice-consuming countries [25]. This demographic shift has diversified quality demands, complicating the definition of quality priorities for breeding and adaptation efforts. The industry maintains a significant export orientation of short- to medium-grain *Japonica* type, with 2022–2023 export values of AUD 394 million against imports of long-grain Basmati and Jasmine quality type accounting AUD 312 million, indicating both strong domestic demand for imported varieties and important international markets for Australian production [23].

Market projections suggest moderate growth, with market size expected to reach USD 172 million in 2024 and maintain a 1.4% annual growth rate through 2028 [26]. However, these projections incorporate significant uncertainty regarding climate change impacts on production capacity, quality parameters, and international competitiveness. The industry’s ability to maintain yield and quality attributes under changing climate conditions will directly influence its position in premium market segments.

### 2.3. Climate Vulnerability and Adaptation Capacity

The contemporary Australian rice industry demonstrates specific climate vulnerabilities and adaptation capacities that will determine its sustainability under projected climate scenarios. Key vulnerabilities include (1) water dependency in a region experiencing increasing precipitation variability and competing water demands; (2) temperature sensitivity during critical reproductive and grain filling stages; (3) geographical concentration, which creates systemic vulnerability to localised climate impacts; and (4) a quality differentiation strategy, which requires maintenance of specific parameters under changing conditions [27,28,29].

The increasing frequency and severity of extreme weather events, including prolonged droughts and floods, and raising temperatures overlapping with grain set and filling presents adaptation challenges that exceed historical experience. The devastating 2019–2020 bushfires exemplify these impacts, while major agricultural areas avoided direct fire damage, associated drought conditions severely reduced crop yields, with irrigation storages depleted to critically low levels [30].

This profile of specific vulnerabilities and adaptation capacities creates a complex adaptation landscape requiring integrated approaches spanning genetic, agronomic, technological, and institutional domains. The industry’s historical experience with environmental constraints provides valuable adaptation foundations, though projected climate change rates may exceed historical adaptive capacity without accelerated innovation.

To counterbalance these vulnerabilities, the Australian rice industry needs to adapt holistic approaches, which includes significant adaptation capacities including (1) established breeding programmes focused on environmental stress tolerance; (2) advanced water management systems with demonstrated efficiency improvements; (3) technological innovation capacity supporting precision agriculture approaches; and (4) institutional knowledge systems facilitating information transfer and coordinated response with strengthened value chain.

## 3. Global Temperate Rice-Growing Regions and Rice Grain Quality Classes

Temperate rice breeding programmes worldwide have developed region-specific strategies to maintain and enhance quality segmentation, responding to the complex interplay between consumer preferences, environmental constraints, and market demands. These strategies have produced benchmark varieties that define the quality standards for temperate *japonica* rice across diverse production regions (Table 1). These benchmark varieties serve multiple critical functions: they establish reference points for quality assessment, create targets for breeding programmes, and provide genetic resources for climate adaptation efforts. The establishment of these quality benchmarks reflects each region’s distinct culinary traditions, consumer preferences, and environmental conditions, creating a complex matrix of quality parameters that must be maintained under changing climate conditions.

In temperate East Asia, varieties such as Koshihikari (Japan) and Dongjinbyeo (Korea) define quality standards emphasising soft texture, low amylose content, and glossy appearance [53,54]. These quality characteristics reflect deep cultural preferences and showcase the profound influence of consumer demand on breeding objectives in these regions. The primacy of texture and appearance in East Asian quality assessment frameworks has significant implications for climate adaptation efforts, as these parameters demonstrate sensitivity to temperature fluctuations during grain development [55,56].

European temperate rice production, centred in Mediterranean countries including Italy, Spain, and France, has established distinct quality profiles oriented toward culinary applications like risotto. Benchmark varieties, such as Arborio and Carnaroli in Italy as well as Bomba and Bahia in Spain, define quality standards characterised by specific textural properties and cooking behaviour [45,46]. These varieties exhibit higher amylopectin-to-amylose ratio that produces the creamy texture and mouthfeel essential for traditional European rice dishes, rendering them vulnerable to climate factors that affect grain filling, starch composition and granule structure.

The California rice industry thrives in the temperate climate of the Sacramento Valley, where hot days and cool nights create ideal conditions for growing high-quality *japonica* rice. More than 85% of all California rice is Calrose, a medium-grain variety prized for its soft, sticky texture and recognised both nationally and internationally. About 10% of rice grown in California consists of short-grain varieties (Koshihikari and Akitakomachi). These varieties are valued for their ideal texture and cooking properties, particularly in sushi and other Asian cuisines [57,58].

Temperate rice production extends to regions not traditionally associated with rice cultivation, including Eastern Europe, Russia, and temperate regions of South America. In these areas, benchmark varieties emphasise adaptation to local environmental conditions alongside quality parameters, representing a different balance between environmental resilience and quality maintenance.

This diversity in quality benchmarks across temperate regions creates both challenges and opportunities for climate adaptation, as it provides genetic resources with varied adaptive capacities while requiring maintenance of diverse quality profiles under variable climatic conditions [59,60].

### Quality Class Differentiation and Climate Vulnerability

Temperate rice-growing regions cultivate a diverse range of rice quality classes, each presenting distinct climate vulnerabilities related to their biochemical composition, genetic background, and quality parameters. Medium-grain *japonica* varieties constitute the dominant quality class in many temperate regions, valued for their soft, slightly sticky texture resulting from specific amylose–amylopectin ratios and protein profiles. These varieties serve as the foundation for dishes ranging from steamed rice to rice-based desserts, with their multi-purpose culinary applications creating significant economic and cultural importance [12,13,49]. Their quality profile demonstrates specific vulnerabilities to temperature fluctuations during grain filling, which can alter starch structure and composition with consequent effects on cooking behaviour and textural properties.

Short-grain rice varieties, including those used for sushi production, share fundamental characteristics with medium-grain types but exhibit even stickier texture due to low amylose preferences with specific starch structural properties. Koshihikari represents the definitive benchmark for this quality class, with its soft characteristic texture, glossy endosperm, and minimal chalkiness establishing quality standards across major temperate rice-growing countries [39]. The exacting quality requirements for these varieties create climate vulnerabilities, as the biochemical pathways governing starch biosynthesis demonstrate sensitivity to temperature fluctuations and water stress during critical developmental periods. Elevated night temperatures have been shown to substantially increase percent chalk and reduce head rice yield as well as affect starch metabolism by triggering alpha amylases [1,61].

While traditionally associated with tropical climates, jasmine rice is increasingly cultivated in some temperate regions where aromatic rice demand exists. Premium *indica* varieties classified under the jasmine rice quality class, such as Khao Dawk Mali 105 (KDML105), serve niche markets with distinct preferences for fragrant aromas and soft texture [43,62]. These aromatic varieties exhibit specific climate vulnerabilities related to the biosynthetic pathways governing aroma compound production, which demonstrate sensitivity to temperature and water availability during grain development [63].

Italian rice production represents a specialised segment within temperate *japonica* cultivation, with varieties such as Carnaroli, Arborio, Roma, Baldo, Vialone Nano, Sant Andrea, and Ribe developed for specific culinary applications. Italy’s prominence in rice production is reflected in the number of varieties recognised under the European Union’s quality schemes, including Protected Designation of Origin (PDO) and Protected Geographical Indication (PGI). These labels distinguish agricultural products with specific regional characteristics and high-quality standards. Among the PDO-designated rice types are those from the Baraggia Biellese and Vercellese regions, including varieties such as Arborio, Baldo, Balilla, Carnaroli, Sant Andrea, Loto, and Gladio. PGI recognition applies to Veronese rice (notably Vialone Nano) and Po Delta rice, which includes cultivars like Arborio, Baldo, Carnaroli, Volano, Telemaco, Cammeo, Karnak, Caravaggio, and Keope. These rice types are valued for their distinct regional identity, traditional cultivation techniques, and superior sensory qualities, ensuring a consistently high standard in flavour and texture for consumers [45,64].

Non-fragrant long-grain rice varieties grown in temperate regions serve consumers preferring drier, separate-grained texture resulting from higher amylose content and specific protein characteristics. These varieties demonstrate different climate vulnerabilities compared to medium-grain types, particularly regarding temperature sensitivity during starch synthesis phases. Basmati rice, traditionally grown in subtropical regions of South Asia, presents an interesting case where genomic studies indicate closer genetic similarities with *japonica* than with typical *indica* varieties despite its classification [50,65,66]. This genomic complexity may influence its climate adaptation capacity in temperate regions and offers potential genetic resources for climate resilience breeding.

## 4. Impact of Climate Change and Fluctuating Environmental Conditions on Grain Quality

Climate change is fundamentally altering the environmental conditions under which temperate rice is cultivated, with significant implications for both yield stability and grain quality parameters. In temperate regions, these environmental shifts pose challenges due to their specific effects on the biochemical and physiological processes governing quality traits in *japonica* rice varieties which will be explored further in the last part of this section. The combined influence of temperature extremes, altered precipitation patterns, elevated atmospheric CO_2_, and increased salinity creates a complex matrix of interacting stressors that affect grain development and quality characteristics through multiple pathways. Beyond direct climatic influences, shifting pest and disease dynamics, soil deterioration, and changes in water resource availability further complicate rice production in temperate regions. Understanding these complex interactions is essential for predicting long-term impacts on *japonica* rice quality and developing targeted adaptation strategies, including climate-resilient varieties and optimised agronomic practices [67,68,69]. In Australian production systems, timing cultivation to ensure vegetative growth during warmer temperatures and high solar radiation, while positioning grain filling during milder autumn temperatures, represents a key adaptive strategy for quality management [23,49,70]. This approach offers transferable lessons for other temperate rice-growing regions facing similar climatic challenges.

The interaction between grain quality parameters and climate factors in temperate rice production creates a complex adaptive challenge requiring integrated approaches spanning genetics, agronomy, and technological innovation. Temperature regimes during grain development exert particularly significant influences on quality formation, affecting enzymatic activities governing starch biosynthesis, protein accumulation, and aroma compound production. Cold temperature stress during critical reproductive stages represents a distinct challenge in temperate systems, with potential effects on grain filling, starch structure, and ultimately cooking quality [7,69,71].

Consumer preferences regarding quality characteristics demonstrate substantial regional variation while exhibiting general resistance to change, creating adaptation challenges where climate impacts affect established quality parameters. This fixed consumer preference creates imperative for breeding and management approaches that maintain the consistency of quality under changing environmental conditions, rather than attempting to shift consumer expectations to accommodate climate-induced quality alterations. This consumer-driven constraint significantly shapes adaptation pathways and priorities across temperate production regions [72].

The differentiation of quality classes across temperate regions provides potential genetic resources for climate adaptation while creating challenges related to maintaining distinct quality profiles under changing conditions. Understanding these quality–climate interactions requires integrated research approaches examining biochemical pathways, genetic regulation, environmental influences, and their combined effects on grain quality and nutrition. This understanding forms the foundation for developing targeted adaptation strategies that address specific vulnerabilities across quality classes while maintaining the distinct characteristics that define market segments and consumer preferences.

### 4.1. Temperature Stresses

Temperature extremes are among the most critical environmental factors affecting rice growth and grain quality in temperate production systems. Heat and cold stresses, when occurring during sensitive developmental stages, cause substantial yield losses and quality deterioration. Heat stress, defined as exposure to temperatures exceeding optimal growth conditions during stages such as booting, flowering, and grain filling, shows regional variation in critical thresholds. A global consensus identifies 33 °C as the critical temperature beyond which rice yields decline, while Australian temperate systems tolerate higher thresholds (35–42 °C), depending on developmental stage and exposure duration. This increased tolerance is likely due to lower humidity levels that facilitate more efficient transpiration-mediated cooling [73,74,75,76].

Conversely, cold stress occurs when temperatures drop below 15 °C during reproductive stages, particularly affecting panicle initiation and flowering. Temperatures below 10 °C during the young microspore stage can cause severe damage to pollen development and grain formation [77,78,79]. Both heat and cold stresses significantly impair grain quality, but through distinct mechanisms. Heat stress increases the percentage of chalky grains and reduces amylose content and grain weight. Cold stress, on the other hand, reduces spikelet fertility and increases the number of unfilled grains, compromising both yield potential and market value [80,81]. Understanding these temperature–quality relationships is essential for developing effective mitigation strategies that integrate genetic innovation, optimised agronomic practices, and precise crop management. These approaches are vital for ensuring sustainable rice production under increasing climatic variability [40,82].

#### 4.1.1. Cold Temperature Stress

Cold temperature stress represents a distinctive challenge in temperate rice production systems, particularly during critical reproductive and grain-filling stages. Australian research has extensively documented yield impacts from low-temperature stress [7,83], though investigations specifically examining grain quality implications remain comparatively limited [83,84].

Low temperatures during key developmental phases can reduce rice yields by over 40%, with cold snaps (<15 °C) during pollination proving particularly damaging by inducing spikelet sterility and disrupting fertilisation processes. The development of cold-tolerant varieties like Sherpa demonstrates the Australian industry’s adaptive response to this challenge. This variety provides enhanced resilience by requiring less water to protect against low temperatures while maintaining yield stability through critical growth stages [85]. The physiological mechanisms underlying cold tolerance include improved pollen viability under low temperatures and enhanced capacity to maintain metabolic processes during stress periods. These adaptations extend to flowering and grain-filling phases, reducing cold-induced sterility and stabilising yield potential. Additionally, shorter growth cycles help ensure rice matures before potentially damaging cold weather events, mostly in February, that coincide with critical reproductive growth stages [6,86,87]. In the Riverina, there’s a 50% probability of mean night temperatures falling below 15 °C for 10 consecutive days during late February, which coincides with the flowering stage, one of the most cold-sensitive periods in rice development [77].

Low temperatures during early developmental stages can indirectly affect grain quality by altering subsequent grain-filling processes [88]. Research from southern China, experiencing more frequent low temperatures and reduced solar radiation during heading and grain-filling stages, demonstrates how these conditions negatively impact multiple quality parameters. The combination of low temperature and reduced light intensity increases chalkiness incidence and severity while reducing milling quality, which significantly impacts head rice yield. Additionally, these stress conditions alter starch viscosity properties, with the most pronounced effects occurring during the initial 21 days of grain filling [89].

The Riverina region in southeastern Australia benefits from abundant solar radiation and typically cooler grain-filling periods that generally favour enhanced quality. However, when low temperatures during reproductive development coincide with other stressors, yield potential and quality parameters face compound threats [90]. This regional experience highlights the importance of expanded research on rice grain quality responses to temperature fluctuations in Australian production systems, particularly during autumn when similar environmental conditions affect grain-filling processes.

#### 4.1.2. High Temperature Stress

Rice cultivation requires temperatures above 20 °C but below 35–40 °C, with the optimal range for growth lying between 20 and 30 °C [91]. High-temperature stress during grain development significantly impairs grain quality by shortening the grain-filling period, reducing grain weight, and increasing the incidence of chalkiness and spikelet sterility. Elevated temperatures interfere with critical physiological processes involved in starch biosynthesis, aroma production, and grain structural integrity, leading to declines in head rice yield, cooking quality, and overall market value. Elevated temperatures (>30 °C) during grain filling disrupt the expression and activity of granule-bound starch synthase I (GBSSI) enzyme, directly reducing amylose content and altering pasting properties [31,80]. Aromatic varieties possess unique vulnerability related to 2-acetyl-1-pyrroline (2AP) accumulation pathways. Heat stress downregulates key genes in the 2AP biosynthetic pathway, including betaine aldehyde dehydrogenase (BADH2) enzyme, directly reducing the synthesis of this fragrance compound [41]. Paradoxically, moderate drought and salinity stress can enhance 2AP concentration through increased proline accumulation, which serves as a precursor for 2AP synthesis, demonstrating how the severity of different environmental stressors can produce opposing effects on specific quality attributes [40]. However, it must be noted that during high temperatures, the volatile compounds synthesized during grain development can potentially evaporate, thus retaining aroma under high temperature is a difficult challenge.

Anthropogenic climate change has significantly increased environmental temperatures in many regions, negatively affecting both crop productivity and grain quality parameters. High-temperature stress during grain development reduces grain filling duration and negatively impacts grain quality, increases chalkiness incidence and severity, lowers head rice yield, and lowers grain weight in temperate rice varieties, directly affecting multiple quality attributes [90]. Importantly, research distinguishes between HNT effects and high day temperature impacts, with HNT demonstrating particularly significant influences on grain quality impairment.

It has been found that HNT (22 °C/34 °C night/day) reduces final grain weight by slowing grain growth during early to mid-grain filling. This occurs partly due to reduced endosperm cell expansion, especially between the central and peripheral zones of the endosperm. High day temperatures (HDT; 34 °C/22 °C) have a lesser effect, underscoring the greater impact of nighttime heat stress [92]. HNTs (26–27 °C) reduce grain yield by 8–17% compared to low night temperatures (22–23 °C), primarily due to decreased biomass accumulation, reduced crop growth rates, and lower harvest indices. At the grain level, HNT negatively affects grain weight, shortens grain filling duration, and reduces spikelet fertility, collectively contributing to the observed yield losses [93].

The cellular and molecular mechanisms underlying HNT effects include activation of stress signalling pathways that can cause permanent cell membrane damage. These disruptions to cellular integrity manifest as reduced kernel quality and increased chalkiness, directly lowering market value [91]. Research in southern China’s double rice cropping system found that post-anthesis warming increased canopy temperatures without significantly affecting grain yield. However, this elevated temperature reduced milling and appearance quality while paradoxically improving eating and nutritional quality parameters, demonstrating the complex and sometimes contradictory effects of temperature on different quality attributes [94].

According to NSW DPI and RGA, timing the cultivation to ensure vegetative growth during warmer temperatures and high solar radiation, as well as positioning grain filling during milder autumn temperatures, represent key adaptive strategies for quality management in the Australian rice production system [70,95]. Australian research examining heat stress during different developmental stages found significant yield losses when high temperatures coincided with panicle exertion and early grain filling, though genetic variation in tolerance was evident [96]. Varieties including YRM 67, Koshihikari, and Opus demonstrated greater heat tolerance during these phases. During late grain filling, yield reductions appeared in specific varieties (Opus and YRM 67) while others remained relatively unaffected. Though genetic variation exists, no variety demonstrated consistent heat tolerance across all developmental stages, highlighting the need for broader germplasm screening and targeted breeding efforts [96].

### 4.2. Water-Related Stresses and Management

Water availability represents a fundamental constraint in temperate rice production systems, encompassing three major stress types: water stress (insufficient water for optimal physiological function), drought stress (severe soil moisture deficit that prevents completion of the life cycle), and salinity stress (excessive salt concentrations that impair growth and fertility) [97,98,99]. In response to these challenges, the Australian rice industry has identified quantitative water management thresholds. For example, soil moisture deficits exceeding 15 kPa have been shown to delay panicle initiation and reduce yield by up to 55% [100]. Water-saving irrigation strategies, such as Alternative Wetting and Drying (AWD), have been shown to reduce irrigation water use by up to 35% compared to continuous flooding, while maintaining comparable grain yields in rice production [101,102]. Additionally, Delayed Permanent Water (DPW) practices improve water productivity, enhancing the efficiency of water use in Australian systems [103]. These adaptive irrigation strategies underpin Australia’s internationally recognised water use efficiency and offer valuable insights for other temperate rice-growing regions facing intensifying hydrological pressures under climate change.

#### 4.2.1. Altered Precipitation Patterns and Water Management Implications

Changes in precipitation patterns directly affect water availability for irrigation while creating secondary effects through humidity levels, disease pressure, and soil conditions. Research examining rice responses to different seasonal conditions found that grain yield and quality parameters, including head rice yield, chalkiness incidence, and gelatinisation temperature, responded differently to wet, cool, and hot seasonal conditions. These findings indicate that adaptation strategies must consider both yield stability and grain quality maintenance under increasingly variable climate conditions [104].

Water management practices interact with temperature effects to influence quality development, creating complex vulnerabilities in temperate rice production systems. The Australian rice industry exemplifies this interaction, having developed sophisticated water management approaches that address both resource efficiency and quality maintenance objectives [70,83,105,106]. These management approaches must continually evolve to address changing climate conditions, particularly increasing temperature variability and altered precipitation patterns that affect both water availability and grain quality.

Direct-Seeded Rice (DSR) is increasingly recognised as a resource-conserving technology that reduces water use and labour input while enabling mechanisation and early crop establishment. When combined with AWD (Section 6.2 in this review) irrigation, DSR enhances water use efficiency and water productivity, offering a sustainable pathway for rice production under limited resource availability [107]. In addition to water savings, AWD with DSR have been shown to significantly reduce greenhouse gas emissions—particularly methane (CH_4_) by 60–87%—and lower total arsenic concentrations in milled rice grain by up to 65%, without compromising grain yield or requiring increased nitrogen inputs. This reduction in grain arsenic levels represents a notable improvement in rice grain quality from a food safety perspective, reinforcing AWD’s role as a climate-smart and quality-conscious irrigation strategy in rice systems [108].

Research from Charles Sturt University has demonstrated that water-saving irrigation practices can maintain key quality parameters when properly managed. Wood et al. [83] found that DPW with post-flower flush supplementation maintains milling quality when combined with appropriate nitrogen management (>60 kg/ha), though the effects vary by variety and quality class. Interestingly, some water stress during the vegetative growth phase can actually improve grain quality during subsequent grain-filling periods through physiological adaptations that enhance translocation efficiency [109]. These findings highlight the complex interactions between water management strategies and grain quality formation. Further, deploying drought-tolerant and water-efficient rice varieties can help mitigate the impacts of altered precipitation patterns. These varieties are bred to withstand drought conditions and use water more efficiently, which is crucial in temperate production zones facing climate change [6].

Precipitation changes affect both direct water availability for irrigation and broader hydrological systems supporting rice cultivation [110]. Increased humidity from higher rainfall can promote disease development, indirectly affecting both yield potential and quality parameters, particularly milling quality. Excessive rainfall sometimes causes waterlogging issues, though rice demonstrates better tolerance to saturated conditions than many other broadacre crops. This was evident in Australia’s Northern Rivers region, where rice crops experienced fewer losses than other commodities during extreme flooding in February and March 2022 [23].

Extreme precipitation events during grain filling, particularly those combining increased rainfall with reduced solar radiation, significantly reduce whole grain percentage while increasing the proportion of immature and chalky grains. These quality impacts under wet conditions demonstrate how precipitation extremes can directly impact market value through appearance and milling quality deteriorations [111]. Water management adaptations under changing precipitation regimes must therefore consider quality implications alongside water use efficiency objectives.

#### 4.2.2. Increased Salinity

It is important to distinguish between different types of salinity affecting agricultural systems. While irrigation salinity in broader landscapes often arises from rising water tables and inefficient irrigation practices, salinity in rice cultivation is more commonly associated with the direct use of poor-quality irrigation water, particularly in regions where freshwater resources are limited. In both cases, the diffusion of salt through the soil plays a central role, but the mechanisms and management strategies differ. For rice, the immediate concern is the accumulation of salts in the root zone from saline water, which can impair plant growth, reduce yield, and degrade soil structure over time [99,112,113]. Rice demonstrates high sensitivity to salinity compared to other cereal crops, with most varieties exhibiting a salinity threshold around 3 dS/m [114].

Research examining *japonica* rice under moderate salinity stress (1.11 dS/m) found compromised grain quality compared to control conditions (0.21 dS/m), with reduced amylose content and increased protein content. Interestingly, appearance quality remained relatively unaffected, suggesting differential sensitivity across quality parameters [115]. Studies with *japonica* rice (Nipponbare) under low-to-moderate salinity stress (2 and 4 dS/m) revealed that timing influences response patterns. Salinity positively influenced starch accumulation under seedling and anthesis treatments, while higher salinity levels (4 dS/m) reduced grain weight and seed viability. Protein and nitrogen content increased under these conditions without significantly altering starch fine structure or composition, demonstrating complex and context-dependent responses to mild salinity stress [116].

While literature on salinity effects on grain composition and quality remains limited compared to its impact on yield [99,113,117,118], evidence indicates that salt-induced changes during vegetative development can significantly affect subsequent grain development, composition, and quality formation [119]. Rice demonstrates highest salinity sensitivity between the three-leaf stage and panicle initiation, creating a critical window where salinity management has particularly significant quality implications [113].

Comparative analysis of salt-tolerant and salt-susceptible rice varieties revealed differential quality responses across tolerance groups. Salt-tolerant varieties initially showed improved milling quality and reduced chalkiness at lower salinity levels, with these benefits diminishing at higher salinity. Salt-susceptible varieties demonstrated consistent quality degradation as salinity increased. This variable response suggests potential for targeted breeding, utilising salt-tolerant germplasm to maintain quality under increasingly saline conditions [120]. Interestingly, moderate salt stress (0.1% NaCl) improved appearance, milling, and eating qualities in high-quality *japonica* cultivars despite yield reductions. These improvements corresponded with favourable changes in starch structure and physicochemical properties, suggesting potential for cultivating premium rice in mildly saline environments [121].

### 4.3. Elevated Atmospheric CO_2_ Concentrations

Rising atmospheric CO_2_ concentrations, currently around 420 ppm, are projected to reach approximately 550 ppm by 2050 and 700 ppm by 2100. While elevated CO_2_ can enhance photosynthesis and yield in rice, it also poses challenges for maintaining grain quality due to altered nutrient composition and physiological responses [122,123,124]. Elevated CO_2_ concentrations around 550 ppm, as simulated in Free-Air CO_2_ Enrichment (FACE) experiments globally, have been shown to increase rice yields while often reducing grain quality traits such as protein content, mineral nutrient concentrations, and increasing grain chalkiness [123,125]. These observations are consistent with global FACE studies reporting 10–14% reductions in protein and 15–30% decreases in mineral concentrations in non-leguminous crops [126,127]. Interestingly, elevated CO_2_ may positively affect rice cooking and eating quality by enhancing grain stickiness and texture balance while reducing hardness [125].

Increasing atmospheric CO_2_ concentrations present complex implications for rice production, potentially boosting yield while simultaneously altering grain quality parameters. Field experiments using FACE technology demonstrate that elevated CO_2_ can enhance rice yield while deteriorating certain grain quality parameters, particularly milling quality and nutritional attributes [128]. The mechanisms underlying these changes include altered carbon partitioning, modified nitrogen metabolism, and changes in starch biosynthesis pathways.

Research with the short-duration Australian cultivar Jarrah revealed increased yield under elevated CO_2_ conditions, primarily through increased grain number and size. These yield benefits came with significant quality alterations, including firmer cooked rice texture but reduced grain nitrogen and protein concentration. Phosphorus content per grain increased under elevated CO_2_, demonstrating differential effects across nutritional components. These findings highlight the need to develop rice genotypes that maintain quality attributes under rising CO_2_ levels [34], considering the possibility of quality improvement at slightly elevated CO_2_ but potential deterioration in further elevation.

The nutritional implications of elevated CO_2_ extend beyond protein content, with studies showing reduced grain nutrient levels but increased heavy metal concentrations under higher CO_2_ conditions. Canopy warming mitigates some nutrient losses but increases metal accumulation risks when combined with elevated CO_2_ [129]. Comparative analysis of wild and domesticated rice under elevated CO_2_ found increased grain size and weight across genotypes, with altered starch gelatinisation properties showing genotype-specific responses. Nitrogen and amylose content remained relatively stable, indicating differential sensitivity across quality parameters and potential genetic resources for adaptation in wild germplasm [35].

Recent research in China examining *japonica* rice under elevated CO_2_ (550 μmol/mol) in open-field conditions found decreased head rice yield, increased chalky grain percentage, and reduced appearance quality. Paradoxically, cooking and eating quality improved under these conditions, evidenced by altered starch RVA (Rapid Visco Analyzer) profiles and enhanced palatability. However, nutritional quality declined, with reductions in essential nutrients including nitrogen, phosphorus, zinc, and amino acids [125]. Comparative analysis between *japonica* and *indica* cultivars under elevated CO_2_ revealed that *japonica* varieties (Wuyunjing27) exhibited greater quality deterioration than *indica* types (Yangdao6), suggesting potential for exploiting *indica* genetic resources to maintain quality under future CO_2_-enriched conditions [130].

### 4.4. Extreme Weather Events

Temperate rice production in the Riverina region faces growing risks from extreme weather, including heatwaves, drought, flooding, hailstorms, and severe storms. Despite some recent rainfall, drought-affected areas continue to expand, with a 40–65% chance of above-median temperatures and variable rainfall projected for March to May. Early 2025 saw flash floods and hail damage from severe storms, while ongoing floodwaters threaten crops downstream. These challenges highlight the urgent need for adaptive management to protect rice yield and quality under increasingly variable climate conditions [131,132,133].

Historical records highlight rice vulnerability: the 2019–2020 bushfires and drought severely affected production through extreme heat and water scarcity, while the 2022 floods caused significant crop inundation, impacting both quality and yield. However, rice demonstrated greater resilience to waterlogging compared to other broadacre crops [23,134,135]. Climate change projections forecast increased frequency and intensity of such extremes in temperate rice-growing regions, underscoring the need for heat- and drought-tolerant varieties, enhanced forecasting systems, and adaptive management strategies to sustain production under more variable conditions [136,137].

Australian agriculture faces increasing disruption from more frequent extreme weather events, including floods, droughts, and bushfires [138]. The devastating 2019–2020 Australian bushfires exemplify these impacts. While major agricultural areas avoided direct fire damage, associated drought conditions severely reduced crop yields, with irrigation storages depleted to critical levels. These conditions produced the smallest rice crop in Australian production history, demonstrating extreme event impacts on production capacity [30].

Drought conditions severely affect both rice yield and quality parameters [83]. Water limitation during grain development leads to multiple quality defects, including smaller grain dimensions, incomplete grain filling, increased chalkiness, and reduced milling quality. Drought stress often causes grain cracking, directly affecting appearance and market value, while potentially altering biochemical composition, including reduced amylose content affecting cooking quality [139,140].

The increasing frequency and severity of extreme events poses challenges for quality maintenance in temperate rice production. These events often create compound stresses combining temperature extremes, water limitations, and altered radiation conditions that affect multiple quality parameters simultaneously. Developing adaptation strategies addressing these compound stresses requires integrated approaches spanning genetics, agronomy, and water management to maintain quality stability under increasingly variable climate conditions.

To illustrate the regional variation in climate-related challenges, Table 2 summarises the relative severity of key environmental stressors across major temperate rice-growing regions. This comparison highlights both shared vulnerabilities, such as the widespread impact of elevated CO_2_, and region-specific risks that shape adaptation priorities.

### 4.5. Physiological and Molecular Mechanisms Underlying Climate Stress Effects on Rice Quality

Climate stress affects rice grain quality through well-established physiological pathways that modify key biochemical processes during grain development. Understanding these mechanisms provides the foundation for developing targeted adaptation strategies, though the complexity of stress responses means that many regulatory relationships require further experimental validation. Under heat stress (>33 °C), rice plants respond through temperature sensing mechanisms that trigger protective responses including heat shock protein activation. In sensitive *japonica* varieties such as Koshihikari and Calrose, elevated temperatures during grain filling significantly reduce granule-bound starch synthase (GBSS) activity, leading to decreased amylose content and increased grain chalkiness. Research demonstrates that heat stress can significantly reduce amylose content through disruption of starch biosynthesis enzymes, with these effects being particularly pronounced in medium-grain varieties that rely heavily on GBSS for amylose synthesis [141,142].

Drought stress activates well-characterised signalling pathways involving abscisic acid (ABA) accumulation and stress-responsive gene expression. This upregulates nitrogen metabolism genes including glutamine synthetase (GS), which contributes to enhanced protein accumulation during stress. Drought conditions also affect the biosynthesis of aromatic compounds, particularly 2AP in fragrant varieties, though the precise regulatory mechanisms controlling this enhancement remain under investigation. The increase in protein content under drought reflects altered carbon–nitrogen partitioning rather than absolute increases in nitrogen uptake [143,144].

Cold stress (<15 °C) affects rice quality primarily through reduced enzyme kinetics and disrupted cellular membrane function. Low temperatures impair grain filling processes by reducing the efficiency of starch and protein synthesis pathways, leading to shortened grain filling duration and increased spikelet sterility. Cold-tolerant varieties like Sherpa demonstrate enhanced capacity to maintain metabolic processes under low temperatures, though the specific molecular mechanisms underlying this tolerance require further characterisation [79,145].

Salinity stress (50–100 mM NaCl) impacts grain quality through disruption of ion homeostasis and osmotic balance. High-affinity potassium (HKT) transporters play crucial roles in managing sodium exclusion and maintaining cellular ion balance. Salt stress triggers accumulation of compatible solutes and stress-protective compounds, with tolerant varieties showing enhanced capacity for osmotic adjustment and maintenance of metabolic processes under saline conditions [146,147,148].

Elevated atmospheric CO_2_ concentrations affect grain quality through modifications in photosynthetic carbon assimilation and altered carbon–nitrogen relationships. Increased CO_2_ enhances photosynthetic rates but often reduces grain protein content due to dilution effects from enhanced carbohydrate accumulation. These responses vary significantly between *indica* and *japonica* varieties, with *japonica* types generally showing greater sensitivity to CO_2_-induced quality changes [149,150].

Table 3 summarises the current understanding of environmental stress responses, affected pathways, and quality implications, whilst acknowledging the limitations in our knowledge of specific regulatory mechanisms and the need for additional experimental validation of proposed molecular relationships.

The integration of these physiological responses determines overall grain quality under climate stress, though predicting combined effects of multiple stressors remains challenging due to complex interactions between pathways. Recent systems biology approaches suggest potential coordination through regulatory networks, but experimental validation of these proposed relationships requires substantial additional research using controlled conditions and diverse genetic backgrounds.

Understanding these established physiological mechanisms provides the foundation for developing targeted adaptation strategies through breeding programmes that enhance specific stress tolerance pathways whilst maintaining quality attributes valued by consumers and markets. Future research should focus on experimental validation of proposed regulatory relationships and characterisation of variety-specific responses to facilitate more precise adaptation approaches.

### 4.6. Genetic Network and Metabolic Hubs in Rice Stress–Quality Integration

Recent advances in systems biology have shown that rice responses to abiotic stress are governed by complex, multi-hub regulatory networks rather than linear pathways. These networks integrate environmental signals with grain quality formation through interconnected metabolic and signalling cascades [151]. A key regulatory hub is the energy-stress sensor OsSnRK1A, a protein kinase that coordinates energy status with stress responses. OsSnRK1A regulates genes involved in quality-related processes, including OsNADH-GOGAT2 (involved in nitrogen metabolism and yield) and activates OsMYBS1, which in turn regulates α-amylase gene expression under sugar starvation conditions. Through such signalling, OsSnRK1A links energy metabolism, sugar signalling, and grain development, influencing processes such as starch breakdown and growth repression. It also interacts with pathways such as trehalose metabolism via OsTPP7, further connecting energy sensing to quality outcomes [151,152].

The OsONAC transcription factor family represents another major regulatory hub, comprising over 149 members, of which 63 show overlapping expression under abiotic stress. Members like OsONAC066 act as key transcriptional regulators of stress-responsive genes, contributing to the robustness and redundancy of rice stress signalling networks [153,154,155].

The OsSnRK2/SAPK family, particularly OsSAPK4, SAPK5, SAPK7, and SAPK10, plays a central role in ABA-mediated stress signalling. These kinases regulate transcriptional responses that affect flowering time, grain development, and adaptation to drought and salinity. Together with OsSnRK1A, they illustrate how hormonal and environmental cues are integrated to prioritize stress adaptation over growth, a classic growth–defence trade-off that can directly influence grain quality [156,157,158].

These regulatory systems converge on core quality-determining processes. For example, starch biosynthesis is modulated through the stress-responsive transcription of enzyme-encoding genes such as OsAGPL1/3 and OsGBSSI, which influence amylose content and grain appearance. Similarly, nitrogen metabolism is integrated via the glutamine synthetase family, with overexpression studies showing enhanced stress tolerance and improved grain filling [159,160,161].

In aromatic rice varieties, the biosynthesis of 2AP, the key aroma compound, is regulated by stress-sensitive metabolic pathways. Environmental factors such as salt stress have been shown to increase 2AP accumulation, likely through interactions with proline and methylglyoxal metabolism, though the precise regulatory mechanisms remain to be fully elucidated [98,162]. Finally, network topology analyses suggest that these regulatory systems may exhibit scale-free properties, enabling efficient signal transmission between stress sensors and quality-related outputs [163,164]. However, experimental validation of these properties in rice is still limited and represents a key area for future research.

Appendix A presents a conceptual model of the stress–quality integration network, synthesised from the current literature. This visualisation captures the multi-hub architecture connecting environmental stress sensors to biochemical pathways that determine grain quality traits. The model combines literature-reported interactions and computational predictions, and therefore requires rigorous experimental validation, particularly through multi-omics approaches under controlled stress conditions. The quantitative relationships, connection strengths, and hierarchical organisation depicted are hypothetical and must be validated empirically in specific rice varieties and environmental contexts relevant to temperate production systems.

Understanding these network interactions provides a basis for breeding strategies that manipulate specific regulatory nodes while maintaining pathway coordination essential for quality stability under climate stress. Future research should focus on validating these interactions, mapping regulatory hierarchies, and systematically characterising pathway behaviour across genotypes and environments through integrated bioinformatics and experimental approaches.

## 5. Differential Climate Vulnerability Across Rice Quality Classes

While Section 4 outlined the general physiological and biochemical mechanisms by which climate factors impact rice grain quality, these effects are not uniform across all rice types. This section explores the differential vulnerability of major temperate rice quality classes, medium-grain *japonica*, aromatic, and long-grain varieties, to climatic stressors, based on their unique genetic and biochemical profiles. Understanding these differences is critical to prioritising adaptation strategies tailored to each class’s specific climate sensitivity.

Temperate rice-growing regions cultivate diverse quality classes, each demonstrating distinct climate vulnerabilities related to their biochemical composition, genetic background, and quality parameters. Table 1 synthesises the current understanding of benchmark varieties, quality characteristics, and climate vulnerability across quality classes in temperate production regions.

### 5.1. Mechanistic Basis of Differential Climate Responses

As detailed in Section 4, high temperatures, drought, salinity, and elevated CO_2_ can significantly alter starch biosynthesis, aroma compound production, and grain structure. The variation in climate vulnerability across rice quality classes stems from underlying differences in biochemical pathways during grain development, influencing yield and quality. Medium-grain *japonica* varieties demonstrate sensitivity to heat stress during grain development due to temperature effects on starch synthase activity, which regulates amylose synthesis. Heat stress disrupts starch biosynthetic enzymes such as OsSSIIa and OsBEIIb in rice, reducing amylose content and increasing chalkiness, which degrades grain appearance and cooking quality [165]. Medium-grain *japonica* varieties are known to be particularly sensitive to these effects during grain filling. Elevated temperatures cause a marked decrease in spikelet fertility in medium-grain *japonica* rice, with sterility best predicted by combined air temperature and vapor pressure deficit (VPD) models. This highlights the sensitivity of medium-grain *japonica* to heat stress during flowering and key physiological mechanisms affecting yield [166]. *Japonica* rice naturally has low OsSSIIa and OsGBSSI levels, affecting starch structure and quality under heat stress. Changes in these enzymes reduce amylose content and increase chalkiness, contributing to heat sensitivity in medium-grain *japonica* varieties [167]. Critical temperatures for spikelet sterility in medium *japonica* rice during flowering are around 36.5 °C (Akihikari) and 38.5 °C (Koshihikari, short-grain). Sterility is linked to reduced pollen shedding, and is influenced by humidity and wind, with moderate humidity reducing sterility and high wind increasing it [168]. Further, heat stress reduces spikelet fertility by inhibiting pollen tube growth in Nipponbare, medium-grain *japonica*. Auxin and ROS levels in the pistil are key to this process, with auxin treatment restoring fertility under heat stress in sensitive *japonica* varieties [169].

Aromatic rice varieties show reductions in 2AP fragrance due to suppression of OsBADH2 expression under heat stress. Long-grain varieties with high amylose content exhibit distinct responses to temperature fluctuations due to different starch granule architecture and amylose–amylopectin ratios. These varieties typically contain higher proportions of crystalline starch regions, which demonstrate greater susceptibility to structural disruption under temperature and moisture fluctuations, manifesting as increased kernel cracking and reduced milling quality [48]. These biochemical and structural differences underpin the differential climate sensitivities among rice classes, emphasising the need for targeted adaptation strategies.

Molecular studies have primarily focused on medium-grain *japonica* rice, particularly temperate varieties like Nipponbare, due to their economic importance in East Asia, lower genome-wide diversity, and amenability to transformation and genome editing [170]. In contrast, few studies have focused on long-grain *indica*, short-grain tropical *japonica*, and Arborio rice types. This is largely due to regional cultivation differences, greater genetic diversity, and limited availability of standardised research tools for these classes. This uneven research focus highlights the need for broader molecular and physiological studies across diverse rice classes to support climate-resilient breeding and grain quality improvement [171].

### 5.2. Geographical Variation in Quality Response

Quality response to climate stressors demonstrates significant geographical variation across temperate production regions. East Asian production systems have reported more pronounced quality deterioration under heat stress compared to Australian systems, potentially reflecting differences in night temperature profiles during grain filling [54,172]. European cultivation of Arborio-type varieties shows vulnerability to combined heat and drought stress, with quality deterioration more severe than observed in Mediterranean climate zones because European regions often experience more extreme and unpredictable weather patterns, and traditional irrigation practices may not be as adaptable to rapid climatic changes [45,173].

These geographical differences in climate response highlight the importance of regionally specific adaptation strategies that address the quality vulnerabilities expressed under local environmental conditions. They also suggest potential for knowledge transfer between production regions experiencing similar climate challenges, though modified for local cultivars and quality parameters.

### 5.3. Implications for Adaptation Prioritisation

Differential vulnerability across quality classes necessitates class-specific breeding and management strategies. Medium-grain *japonica* breeding programmes should prioritise temperature stability of starch synthesis pathways, while aromatic rice adaptation requires focus on maintaining 2AP accumulation under variable temperature and moisture conditions.

Water management adaptations must consider class-specific responses, as agronomic practices demonstrate differential quality effects across variety types. Similarly, harvest timing and post-harvest management may require quality-class-specific modifications to minimise climate-induced quality deterioration, particularly for premium classes with stringent quality requirements.

## 6. Adaptation Strategies

### 6.1. Genetic Adaptation Approaches

Breeding programmes in temperate regions increasingly prioritise climate resilience traits alongside quality maintenance, recognising that genetic improvement provides fundamental adaptation pathways for quality preservation. Australian varieties such as Sherpa demonstrate enhanced cold tolerance while maintaining quality parameters under stress conditions [25]. These breeding efforts integrate multiple resistance mechanisms to address compounding stressors in temperate environments [47,48].

Modern breeding methodologies accelerate climate adaptation through complementary approaches: (1) Diverse germplasm screening identifies genetic resources with heightened stress tolerance while maintaining quality attributes. Wild relatives and landrace collections provide particularly valuable allelic diversity for resilience traits [87,174]. Genomic selection tools enhance breeding efficiency for complex quality traits under stress conditions. Marker-assisted selection targeting specific quality-associated loci enables more rapid integration of beneficial alleles into elite backgrounds, with higher precision than phenotypic selection alone [175,176]. (2) Multi-environment and multi-season testing networks evaluate genotype × environment interactions affecting quality stability. These networks systematically assess quality maintenance across temperature and moisture gradients to identify varieties demonstrating quality robustness under variable conditions [177]. (3) Integrated abiotic resistance breeding addresses climate-induced shifts in pest and disease pressure that indirectly affect grain quality. Resistance to pathogens like panicle blast becomes increasingly important as climate change alters disease incidence patterns during grain development stages [178].

Priority traits for Australian temperate conditions include enhanced WUE without quality penalties, improved cold tolerance during reproductive development, and maintained grain quality under heat stress. These breeding objectives require sophisticated phenotyping approaches capable of simultaneously evaluating multiple quality parameters under controlled stress conditions.

### 6.2. Water Management Innovations

Water management strategies provide critical adaptation pathways for quality maintenance under altered precipitation patterns and increased competition for water resources. The Australian rice industry has developed internationally recognised water-saving approaches that balance efficiency objectives while maintaining grain quality [21]. Two primary water management innovations demonstrate significant adaptation potential: (1) Alternate Wetting and Drying (AWD) irrigation cycles flooding and drying phases to reduce water consumption while promoting root development. AWD increases starch thermal stability and alters pasting profiles in some varieties while minimally affecting other quality attributes [179]. The practice reduces arsenic accumulation but may increase cadmium levels, creating quality–safety trade-offs and requiring context-specific evaluation [180]. Though AWD typically reduces yield compared to continuous flooding, it generally maintains milling quality—a critical economic parameter [181]. (2) Delayed Permanent Water (DPW) with post-flower flush supplementation maintains milling quality when combined with appropriate nitrogen management (>60 kg/ha). This approach alters grain protein composition, affecting head rice yield and flour pasting properties while conserving water [139]. The quality impact varies by variety, with medium-grain types generally showing better quality maintenance than long-grain varieties under DPW management.

Water-saving approaches such as AWD and DPW have demonstrated that quality–water trade-offs vary significantly across rice variety types and environmental conditions. While these methods retain elements of flooded cultivation, fully aerobic rice production where rice is grown under non-flooded, upland conditions throughout the season offers even greater water-saving potential [182]. However, fully aerobic systems often result in notable reductions in grain quality and yield, particularly in traditionally lowland varieties like Arborio or other *japonica* types, which are not well adapted to dry soil conditions [183]. Integrated farm management systems that account for variety-specific responses—such as adjusting planting dates, variety selection, and nitrogen management—provide the most effective adaptation pathway for maintaining quality while reducing water use [184]. Incorporating fully aerobic production into this spectrum of strategies emphasizes the need for targeted varietal breeding and localised agronomic adjustments.

### 6.3. Technological Adaptation Systems

Technological innovation provides rapidly evolving adaptation pathways for quality preservation through enhanced monitoring, prediction, and precision management capabilities. Four integrated technological systems demonstrate adaptation value: (1) Climate-responsive decision support systems integrate meteorological data with crop models to optimise management decisions affecting quality development. These systems enable adaptive scheduling of irrigation, fertilisation, and harvest operations based on real-time climate conditions and forecasts [185,186]. GPS-guided technologies further enhance implementation precision, allowing rapid adjustment to climate-induced field heterogeneity [187]. (2) Multi-platform sensing networks combine ground, aerial, and satellite monitoring to detect early indicators of climate stress affecting grain yield and quality. Hyperspectral imaging technologies can identify temperature and moisture stress before visible symptoms appear, enabling pre-emptive management adjustments to preserve quality [188]. Soil sensor networks monitoring moisture, temperature, and nutrient status provide complementary data on root zone conditions influencing grain development [189]. (3) Variable-rate application (VRA) systems optimise resource distribution based on field-specific conditions, mitigating climate-induced spatial variability effects on quality development by responding to microclimate variations within the field and allocating water and nutrients to normalise growing conditions. While widely proven in sprinkler- or drip-irrigated systems, their use in conventional flood-irrigated rice remains largely at the research and pilot stage [190]. (4) Automated irrigation infrastructure adjusts water distribution based on real-time evapotranspiration data and climate forecasts. These systems prevent both water stress and excess moisture conditions that compromise quality, maintaining optimal hydration despite reduced rainfall predictability or increased evaporation rates [191].

The integration of these technological systems creates comprehensive adaptation platforms capable of maintaining quality parameters despite increasing climate variability. Their effectiveness depends on continued improvement in climate prediction capabilities and development of quality-specific stress indicators detectable through remote sensing technologies.

### 6.4. Advanced Quality Assessment Methodologies

Novel quality assessment technologies provide essential adaptation tools by enabling rapid evaluation of climate effects on quality parameters and facilitating selection of climate-resilient phenotypes. Four complementary assessment approaches demonstrate value for climate adaptation: (1) Hyperspectral phenotyping platforms rapidly map chemical composition within individual grains using visible and near-infrared wavelengths. These technologies detect internal quality characteristics including protein distribution, chalkiness development, and structural integrity with minimal sample preparation [192,193]. This capability enables identification of varieties maintaining quality under stress and detection of climate-induced quality deterioration before visible symptoms appear. (2) Near-infrared spectroscopy (NIRS) systems provide rapid, non-destructive quality evaluation across multiple parameters simultaneously. NIRS applications have expanded from basic protein assessment to prediction of complex quality traits including chalkiness, head rice yield, grain dimensions, amylose content, and viscosity profiles [194]. These systems enable high-throughput screening of breeding material for quality stability under stress conditions. (3) Machine learning algorithms integrate multi-parameter data to predict quality outcomes with increasing accuracy. Random Tree modelling approaches have demonstrated superior effectiveness for predicting quality parameters from spectral data [195,196], while artificial neural networks reliably predict both biochemical and functional quality attributes simultaneously [197]. These computational approaches enable more sophisticated understanding of climate–quality interactions and identification of resilient phenotypes. (4) Genomic and metabolomic profiling tools identify molecular signatures associated with quality maintenance under stress. DNA barcoding approaches characterise genetic resources for quality stability [198], while metabolomic analysis reveals biochemical pathways maintaining quality despite environmental fluctuations [199]. These molecular techniques accelerate development of climate-resilient varieties with stable quality profiles. The integration of these assessment methodologies with adaptation strategies creates feedback systems capable of continuous improvement. As climate conditions evolve, these technologies provide increasingly precise understanding of quality responses, enabling more targeted adaptation interventions across genetic, agronomic, and technological domains.

Integrating these genetic, agronomic, technological, and assessment strategies provides a comprehensive foundation for climate adaptation in temperate rice systems. However, their successful implementation and scaling depend heavily on supportive policy frameworks that enable innovation, incentivise quality maintenance, and foster collaboration across stakeholders and regions. To advance practical adaptation, targeted field trials and pilot programs are essential. These include AWD irrigation trials to optimise water savings while maintaining grain quality, varietal screening under FACE systems to evaluate performance under elevated CO_2_, and participatory co-design with farmers to tailor grain quality management practices to local conditions (Table 4). Such applied research bridges gaps between experimental findings and real-world application, accelerating climate-resilient adoption in temperate rice systems.

## 7. Policy Implications for Climate Adaptation

While technical adaptation strategies provide essential pathways for maintaining grain quality under changing climatic conditions, policy frameworks significantly influence their implementation and effectiveness. Climate adaptation requires coordinated policy approaches spanning multiple governance levels to address the complex challenges facing temperate rice production systems. Beyond technical solutions, policy decisions play a foundational role in enabling, scaling, and sustaining climate adaptation efforts. Strategic alignment between science, economics, and governance is therefore essential to long-term adaptation success.

### 7.1. Regulatory Frameworks Supporting Adaptation

Climate adaptation policies must be informed by the physiological and molecular mechanisms underlying rice grain quality vulnerabilities described in Section 4 and Section 5. For example, understanding heat-induced spikelet sterility in medium-grain *japonica* and starch disruption in long-grain rice enables targeted water management, breeding, and pest control policies that protect grain yield and quality.

Water policy frameworks are a key regulatory influence on climate adaptation in the rice industry, particularly in water-constrained regions like the Murray–Darling Basin. Drawing on the ‘Rice in the Riverina’ report [24] and the ABARES [11] reviews of rice investment strategies, this analysis highlights how current water recovery policies, though designed for environmental sustainability, have reduced water availability for rice. Integrating climate projections into water allocation planning could improve industry resilience. This may include more flexible carryover provisions for increased climate variability and extreme events. Integrating climate projections and rice developmental stage sensitivity into water allocation ensures irrigation supports critical periods such as flowering and grain filling, when heat and drought stress most impact yield and grain quality.

Regulatory approaches to pesticide and agricultural chemical registration require reform to accommodate changing pest and disease pressures under altered climate conditions. As highlighted in the ‘Fork in the Road’ report by Farmers for Climate Action [138], current approval timeframes often fail to provide timely access to management tools addressing emerging climate-related threats, potentially compromising grain quality through increased biotic stress. Streamlined registration pathways for climate adaptation tools, particularly those with established safety profiles in comparable jurisdictions, could enhance adaptation capacity without compromising risk assessment integrity.

### 7.2. Economic Incentives for Quality-Maintaining Practices

Economic incentives for climate adaptation currently focus primarily on production volume rather than quality maintenance. Payment structures in the Australian rice industry have historically emphasized yield and broad quality classifications rather than specific quality parameters demonstrating climate vulnerability [71]. Reform of payment structures to reward climate-resilient quality maintenance could better align producer incentives with adaptation objectives. This might include premium payments for maintaining amylose content stability or low chalkiness thresholds under stress conditions. Aligning incentives with maintenance of specific quality parameters such as amylose content and chalkiness, traits disrupted by heat stress as detailed in Section 5, encourages growers to adopt climate-resilient practices that sustain market value.

Carbon market mechanisms represent an underutilised economic incentive for quality-enhancing adaptation practices. Rice cultivation practices like AWD and DPW can simultaneously reduce methane emissions while maintaining certain quality parameters [180]. Policy mechanisms that quantify and reward these emission reductions could enhance the economic viability of adaptation practices, particularly where quality–yield trade-offs might otherwise discourage their adoption. Research and development tax incentives currently provide limited support for private sector investment in climate adaptation technologies. Enhanced incentives targeting climate-resilient quality maintenance could accelerate technological innovation, particularly in precision management and phenotyping technologies requiring substantial development investment [177,200].

### 7.3. Research Funding Priorities

Public research funding for rice climate adaptation has largely focused on yield maintenance, with less emphasis on grain quality. The AgriFutures Rice RD&E Plan 2021–2026 prioritises water productivity and yield gains [201], while quality traits receive comparatively less attention. Greater investment in quality-specific adaptation, such as preserving amylose content, reducing glycaemic index and maintaining appearance quality under stress, could improve sustainability and support premium market positioning.

The current model of regional research prioritisation creates coordination challenges for addressing common adaptation needs across temperate production regions. As stated by NSW DPI, enhanced international coordination mechanisms, particularly for pre-competitive research addressing fundamental quality–climate relationships, could improve research efficiency and accelerate adaptation [19].

Research infrastructure funding for climate adaptation facilities, particularly controlled environment chambers capable of simulating future climate conditions, represents a critical limitation for quality-focused adaptation research. As highlighted by the AgriFutures Australia, enhanced investment in these facilities would enable a more sophisticated understanding of biochemical responses to complex climate scenarios, facilitating targeted adaptation strategy development [85]. Investment in research facilities that simulate complex climate scenarios is essential for dissecting the biochemical and physiological responses highlighted in earlier sections, accelerating the development of climate-resilient rice varieties with stable grain quality.

### 7.4. International Cooperation Mechanisms

Knowledge transfer mechanisms between temperate rice production regions currently operate primarily through ad hoc arrangements rather than structured cooperation frameworks. Development of more formalised knowledge exchange programmes focusing specifically on quality maintenance under climate stress could enhance adaptation capability while avoiding redundant research investments [72]. Germplasm exchange restrictions increasingly limit access to genetic resources for climate adaptation. International policy frameworks facilitating responsible germplasm sharing for climate adaptation purposes, while respecting intellectual property considerations, could enhance breeding programme effectiveness across temperate production regions [7]. Technical standards harmonisation for quality assessment methodologies represents an underutilised cooperation opportunity. Convergence of assessment approaches would facilitate more efficient identification of climate-resilient varieties and practices while enhancing market transparency [194].

Towards addressing these goals, the Temperate Rice Research Consortium (TRRC) was established by the International Rice Research Institute (IRRI) to address challenges in temperate rice improvement through a collaborative approach [202]. It focuses on overcoming both biotic and abiotic stresses, enhancing yield potential, improving grain quality and nutrition, and optimising water and nutrient management. The consortium promotes resource-sharing and information exchange among its 18 member countries and partner institutions. This collaboration has led to significant advancements, such as the development of cold-tolerant and high-yielding rice varieties, streamlined SOPs to assess grain quality traits for matching consumer-driven grain quality criteria, and developing low glycaemic index rice and biotic stress resistance.

## 8. Consumer Perspective on Climate-Induced Quality Changes

Climate adaptation strategies must ultimately address consumer expectations regarding grain quality, which will themselves evolve in response to changing market conditions and product attributes. Understanding potential shifts in consumer preferences under climate change scenarios is essential for prioritising adaptation investments and developing effective market positioning strategies. This will be addressed in the next section.

### 8.1. Potential Shifts in Consumer Acceptability Thresholds

Consumers in established rice markets demonstrate relatively stable preference patterns for specific sensory attributes, including texture, flavour, and visual appearance [72]. However, repeated exposure to climate-induced quality variations may gradually shift acceptability thresholds, particularly for parameters like chalkiness that affect visual appearance more than functional cooking properties. Research examining consumer response to quality variations in wine and fruit markets suggests potential for adaptability in sensory preferences following consistent exposure to altered product attributes [203]. Premium market segments targeting traditional rice dishes (e.g., sushi, risotto) may demonstrate greater resistance to shifted acceptability thresholds due to strong cultural expectations regarding specific quality parameters [56]. Adaptation strategies for these market segments may require prioritisation of stability in culturally significant quality attributes rather than accepting threshold shifts. Price sensitivity influences willingness to accept quality variations, with lower-priced market segments demonstrating greater flexibility in quality acceptability thresholds [204,205,206]. Climate adaptation strategies targeting different market segments may therefore require differentiated approaches based on price–quality positioning.

### 8.2. Implications for Market Segmentation

Climate-induced quality changes may necessitate re-evaluation of established market segmentation strategies. Current quality class designations primarily reflect historical consumer preferences rather than climate vulnerability profiles [39]. Integration of climate resilience characteristics into segmentation frameworks could better align market positioning with adaptation capabilities. Product differentiation based on climate adaptation attributes represents an emerging opportunity for market development. Consumer interest in sustainability credentials creates potential for premium positioning of rice varieties and production systems demonstrating climate resilience while maintaining key quality.

Geographic indication frameworks currently emphasize historical production practices rather than adaptive management approaches that maintain quality under changing conditions [45]. Evolution of these frameworks to accommodate climate-adaptive practices while preserving distinctive quality characteristics would enhance market protection for regional production systems.

### 8.3. Communication Strategies Regarding Quality Variations

Transparency regarding climate-induced quality variations represents an emerging challenge for the rice industry. Current consumer education approaches provide limited information regarding environmental influences on quality attributes, potentially creating unrealistic expectations regarding consistency under changing climate conditions [207].

Sensory education programmes for consumers represent an underutilised approach for building acceptance of climate-induced quality variations while maintaining market engagement. Research in wine markets demonstrates that enhanced knowledge of environmental influences on product attributes can increase acceptance of vintage-related variations [208,209]. Communication strategies emphasising climate adaptation efforts may enhance consumer confidence in industry sustainability while building understanding of quality challenges. Integration of climate resilience messaging with quality narratives could transform potential market threats into differentiation opportunities [23].

Retailer engagement represents a critical element of effective quality communication strategies. Current retail staff training typically emphasizes broad quality classifications rather than specific attribute variations or environmental influences [210]. Enhanced training approaches integrating climate–quality relationships could improve consumer guidance regarding product selection and usage.

### 8.4. Network-Informed Quality Consistency

Consumer acceptance of rice varieties developed through genetic network-based breeding approaches may benefit from improved quality consistency under variable environmental conditions. Unlike traditional breeding focusing on single-trait selection, network-informed strategies targeting genetic hubs and metabolic pathways can potentially reduce quality variability across seasons and production environments. This approach addresses consumer preferences for consistent texture, appearance, nutrition, and cooking properties whilst maintaining adaptability to climate stress. Market communication emphasising ‘climate-stable quality’ rather than stress tolerance *per se* may enhance consumer confidence in network-bred varieties, particularly for premium market segments requiring predictable cooking behaviour [211,212,213,214,215].

## 9. Research Priorities for Climate-Resilient Rice Quality

Addressing climate-related grain quality challenges in temperate rice systems requires targeted research focusing on critical knowledge gaps. Priority areas include systematic investigation of combined stress effects on grain quality across different rice classes, as most glasshouse and field studies predominantly examine individual stressors despite production systems facing multiple simultaneous challenges [216,217]. Expanding molecular understanding beyond medium-grain *japonica* varieties to include long-grain, aromatic, and specialty temperate types such as Arborio remains essential, requiring comprehensive physiological characterisation and experimental validation of stress–quality biochemical pathways through enzyme assays and metabolomics analysis [218]. Development of field-applicable phenotyping technologies represents a critical gap, with priorities including hyperspectral imaging for real-time chalkiness detection, portable near-infrared spectroscopy for protein assessment, and integration with high-throughput breeding platforms. Climate-quality prediction models linking environmental variables to specific grain quality parameters require development and validation, whilst socioeconomic research should address market acceptance of climate-adapted varieties and adoption pathways for quality-focused technologies [69]. Experimental validation of proposed regulatory networks through protein–protein interaction studies and systematic pathway analysis using multi-omics approaches under controlled conditions will clarify the role of complex signalling systems in stress–quality integration, though empirical confirmation remains limited for most computational predictions [164].

## 10. Conclusions

This comprehensive analysis demonstrates that climate change fundamentally alters grain quality in temperate rice production through complex interactions between environmental stressors and biochemical processes during grain development. The evidence reveals that maintaining grain quality under changing conditions requires integrated approaches spanning genetic, agronomic, technological, and policy domains rather than isolated interventions.

Temperature regimes during grain development emerge as the most critical factor affecting quality formation, with well-defined thresholds at >33 °C triggering significant amylose reduction in medium-grain *japonica* varieties and <17 °C disrupting grain filling processes. Heat stress demonstrates the most severe impacts, reducing amylose content through granule-bound starch synthase disruption, whilst paradoxically improving protein content through altered carbon–nitrogen partitioning. Drought stress shows complex effects, enhancing protein accumulation and aromatic compound synthesis in some varieties whilst compromising overall grain filling. Salt stress produces moderate but consistent quality decline through ion transport disruption, though some tolerant varieties maintain quality parameters under mild salinity.

The differential vulnerability across rice quality classes necessitates targeted adaptation approaches. Medium-grain *japonica* varieties require breeding strategies focused on heat-stable starch biosynthesis pathways, whilst aromatic varieties need attention to maintain 2AP synthesis under variable moisture conditions. Long-grain varieties demonstrate vulnerability to combined stressors affecting kernel integrity and milling quality.

The Australian rice industry exemplifies effective adaptation through integrated approaches combining genetic improvement, precision agriculture, and water-efficient practices. The development of cold-tolerant varieties like Sherpa, alongside implementation of water-saving practices such as AWD, demonstrates the potential for maintaining quality whilst adapting to resource constraints. However, the industry’s emphasis on premium quality segments creates vulnerability where climate-induced modifications could undermine market positioning unless adaptation strategies specifically target quality maintenance.

Recent advances in understanding stress–quality relationships suggest potential roles for complex regulatory networks involving multiple signalling pathways, though significant uncertainty remains regarding the precise mechanisms and their practical applications. Current evidence indicates that stress responses operate through coordinated systems rather than simple linear pathways, providing opportunities for targeted interventions through breeding and management practices. However, experimental validation of proposed network relationships and quantification of their practical significance require substantial further research.

Technological innovations provide rapidly evolving adaptation pathways through enhanced monitoring, precision management, and accelerated phenotyping capabilities. Hyperspectral imaging technologies enable early detection of climate-induced quality changes, whilst machine learning approaches improve prediction accuracy for complex quality parameters. These technological advances, combined with improved understanding of physiological responses, create opportunities for more sophisticated adaptation strategies.

Policy frameworks significantly influence adaptation implementation and effectiveness, particularly through water allocation policies, research funding priorities, and economic incentives. Current policies often emphasise production volume over quality maintenance, creating potential misalignment with market positioning in premium segments. Reform of policy structures to better support quality-focused adaptation could enhance industry resilience.

The integration of physiological understanding with practical adaptation approaches remains essential for developing climate-resilient temperate rice systems. Future research priorities should focus on experimental validation of stress–quality relationships, development of robust phenotyping methodologies for field applications, and systematic evaluation of adaptation strategies across diverse production environments. International cooperation mechanisms, particularly through initiatives like the Temperate Rice Research Consortium, represent valuable opportunities for accelerating adaptation knowledge and resource sharing.

The challenge of maintaining grain quality under climate change ultimately requires coordinated efforts spanning multiple scales from molecular understanding to international cooperation. The evidence-based framework established through this analysis provides the foundation for developing targeted adaptation strategies that can sustain high-quality temperate rice production whilst adapting to increasing environmental variability.

## Figures and Tables

**Figure 1 biology-14-00801-f001:**
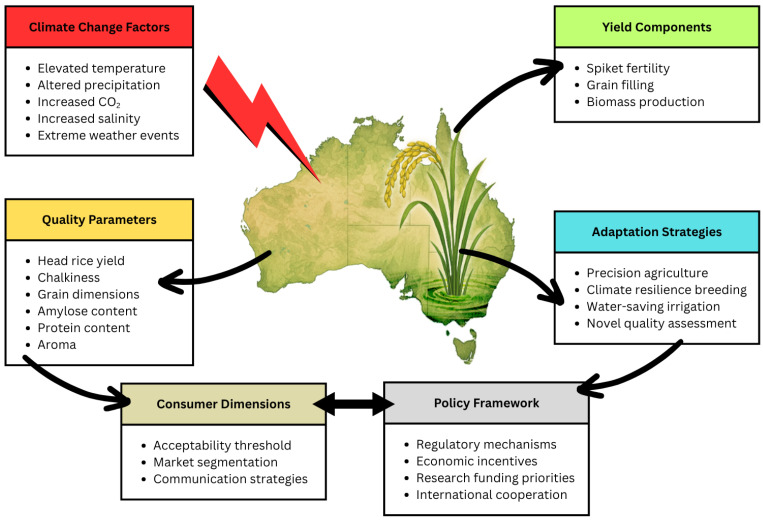
Conceptual framework of climate change impacts on temperate rice production in Australia and global temperate rice growing regions.

**Table 1 biology-14-00801-t001:** Rice quality classes in temperate regions: benchmark varieties, quality parameters, and climate vulnerability.

Quality Class	Region/Countries	Benchmark Varieties	Quality Parameters	Climate Vulnerability	Cooking/Eating Quality Response	Adaptive Practices	Reference
Medium-Grain *Japonica*	Temperate East Asia (Japan, China, and Korea), Australia, and USA (California)	Koshihikari (Japan), Reiziq (Australia), and Calrose (USA)	Soft texture, low-moderate amylose (16–18%), glossy appearance, and good milling quality	Heat stress: decreased amylose content, increased chalkiness, altered crystallinity and gelatinisation temperature, and increased protein content Elevated CO_2_: increased yield, decreased protein and quality	Heat stress produces stickier, softer rice; increased protein potentially reduces stickiness and increases grain hardness	Precision irrigation methods to save water while maintaining milling quality Adjusting planting dates to avoid critical grain-filling during heat peaks Use of cold- and heat-tolerant varieties (e.g., Sherpa)	[4,31,32,33,34,35]
Short Grain (Sushi Rice)	Japan, Korea, and Australia	Koshihikari (Japan), Opus (Australia)	Very low amylose (15–16%), high stickiness, and glossy appearance	Heat stress: increased chalkiness, grain cracking, and protein content; reduced grain size and amylose and starch contents. Elevated CO_2_: increased yield, decreased protein, and increased chalkiness	Texture becomes inconsistent; eating quality in Koshihikari improves under moderate stress but deteriorates under severe stress	Proper irrigation with nitrogen management to improve grain quality Varietal screening and selection for heat tolerance Crop shading or protective structures during heat waves	[36,37,38,39]
Aromatic Rice (Jasmine Type)	Thailand, Australia, and USA	Thai Hom Mali, KDML105 (Thailand), and Topaz (Australia)	Medium amylose (17–19%), distinctive aroma (2AP), and soft texture	Heat stress: reduced 2-acetyl-1-pyrroline production, increased chalkiness Drought/salinity: increased 2AP concentration but reduced yield	Heat stress causes loss of characteristic fragrance; moderate salinity stress can enhance aroma while reducing other quality parameters	Integrated pest and salinity management to maintain aroma and yield Breeding for stable 2AP production under drought/heat Use of microclimate management, e.g., water timing, mulching	[40,41,42,43]
Arborio (Risotto)	Italy, Australia	Arborio, Carnaroli (Italy), and Vialone (Australia)	High amylopectin, intermediate amylose (18–19%), chalky centre, and ability to maintain a firm, *al dente* core when cooked, releasing enough starch to create a creamy risotto	Less tolerant to combined stressors, particularly vulnerable to heat during grain filling	Deterioration in distinctive creamy consistency and texture essential for risotto preparation	Avoidance of combined stresses through irrigation scheduling Selection of varieties with better heat resilience during grain fill Localised nitrogen and water management to maintain texture	[44,45,46]
Non-Fragrant Long Grain	USA, Australia, and Temperate Eastern Europe	Wells (USA), Doongara (Australia), and Rapan (Russia)	High amylose (22–25%), separate grains when cooked, and firm texture, some have low glycaemic index	Heat/water stress: increased chalkiness, reduced grain dimensions Salinity: decreased amylose content	Dry, separate grain characteristics may be compromised; increased stickiness under salinity stress	Management of salinity and water stress through improved drainage and irrigation control Use of drought-tolerant varieties Precision nutrient application to mitigate quality loss	[47,48,49]
Basmati	Northern India, Pakistan, and Australia	Pusa Basmati 1121, Basmati (India/Pakistan), and Basmati Signature (Australia)	High amylose (22-24%), distinctive aroma (2AP), and exceptional kernel elongation during cooking	Temperature fluctuations affect elongation and aroma; high temperature shortens grain filling and reduces starch and amylose contents	Reduced aroma and grain elongation; compromised fluffiness and grain separation valued in premium markets	Timing planting to avoid heat stress during grain fill Breeding for aroma retention and grain elongation under temperature fluctuations Adoption of integrated crop management systems	[37,50,51,52]

**Table 2 biology-14-00801-t002:** Regional comparison of climate stressors affecting temperate rice production systems.

Stressor	Australia	East Asia	Europe	California
Heat stress (>35 °C)	High	Medium	Low	High
Cold stress (<15 °C)	Medium	High	Medium	Low
Water limitation	High	Low	Medium	Medium
Salinity stress	Medium	Low	Low	Low
CO_2_ effects	High	High	High	High

**Table 3 biology-14-00801-t003:** Summary of environmental stressors, molecular response pathways, and adaptation strategies affecting rice grain quality in temperate systems.

Stress Type	Primary Sensors	Key Pathways Affected	Quality Impact	Sensitive Varieties	Adaptation Strategy	Reference
Heat (>35 °C)	Heat shock protein activation, membrane stabilisation	GBSS activity reduction, starch biosynthesis disruption	Amylose ↓, Chalk ↑	Koshihikari, Calrose, medium-grain *japonica*	Heat-tolerant varieties, adjusted planting dates	[141,142]
Drought	ABA accumulation, osmotic adjustment	Enhanced nitrogen metabolism (GS upregulation), altered 2AP biosynthesis	Protein ↑, Aroma ↑	Drought-sensitive varieties, medium-grain *japonica*	Water-efficient varieties, AWD, drought-tolerant varieties	[143,144]
Cold (<15 °C)	Membrane adaptation, enzyme kinetics modification	Reduced metabolic rates, impaired grain filling	Filling duration ↓, grain weight ↓	Cold-sensitive varieties, tropical *japonica*	Cold-tolerant varieties (e.g., Sherpa)	[79,145]
Salinity (50–100 mM)	Ion transport regulation, osmotic balance maintenance	HKT transporter activity, compatible solute accumulation	Variable response depending on tolerance level	Salt-sensitive > tolerant varieties	Salt-tolerant germplasm, improved drainage	[146,147,148]
Elevated CO_2_	Enhanced photosynthesis, altered C:N balance	Modified carbon partitioning, reduced nitrogen concentration	Grain size↑, protein↓	*Japonica* > *indica* sensitivity	CO_2_-responsive breeding targets	[149,150]

Note: Quality impacts and physiological responses show significant variation among varieties and environmental conditions. Mechanisms listed represent well-documented responses, though the precise molecular regulation of many pathways requires further experimental validation. Directional trends in the Quality Impact column indicate: ↑ = increased content or effect; ↓ = decreased content or effect.

**Table 4 biology-14-00801-t004:** Summary of advanced phenotyping and assessment tools for climate-resilient grain quality in temperate rice systems.

Tool/Approach	Description	Climate Relevance/Function	Reference
Hyperspectral Imaging	Uses visible and NIR wavelengths to detect internal grain traits like protein and chalkiness	Enables early detection of stress-induced quality deterioration and identification of resilient phenotypes	[192,193]
Near-Infrared Spectroscopy (NIRS)	Rapid, non-destructive multi-trait analysis	Evaluates amylose content, grain dimensions, chalkiness, and head rice yield under stress conditions	[193,194]
Machine Learning Algorithms	Data-driven models (e.g., Random Tree, ANN) predict multiple quality traits from spectral data	Increases predictive accuracy of grain quality responses to combined stressors	[195,196,197]
Genomic Profiling	Identifies quality-related loci and barcodes climate-resilient varieties	Accelerates breeding for stable grain quality under stress	[198]
Metabolomic Profiling	Measures biochemical pathways and stress-response metabolites	Links physiological mechanisms to quality maintenance across environments	[199]

## Data Availability

No new data were created or analysed in this study.

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
