# Peer review of "Climate Adaptation Strategies for Maintaining Rice Grain Quality in Temperate Regions"

_biology, 2025, doi:10.3390/biology14070801_

Round 1
Reviewer 1 Report
Comments and Suggestions for Authors
REVIEW
On the article Comparative Analysis of Rice Grain Quality and Climate Change Impacts in Temperate Production Zones from an Australian Rice Industry Perspective with authors Yvonne Fernando, Ben Ovenden, Nese Sreenivasulu and Vito Butardo Jr proposed for publication in MDPI – Biology
Climate change and impact on agricultural production and the possibilities for adaptation of agricultural production to them are the subject of numerous scientific studies. The problem considered in this review is part of this general issue and concerns the sustainable production of rice - a major crop related to the nutrition of the population. The article reviews a number of aspects related to the cultivation of rice in temperate latitudes in the context of climatic changes. The introduction systematically and convincingly argues the importance of the impact of climate change on temperate rice grain quality. The primary importance of agroclimatic conditions for the successful cultivation of quality agricultural production and in particular rice is correctly emphasized. The goal and objectives are clearly formulated by the authors. It is noted that climate change is a fundamental challenge for rice production systems in the temperate zone. The role of the integrated approach as a method to adapt to these changes is emphasized. It includes genetic, agronomic and technological areas and areas of assessment. I consider it correct to recommend that Effective Climate Adaptation should extend beyond technical approaches and encompass policy decisions and strategies. The analysis of Global Temperate Rice-Growing Regions and Rice Grain Quality Classes and Quality Class Differentiation and Climate Vulnerability is very useful. I also highly appreciate the detailed analysis of Impact of Climate Change and Fluctuating Environmental Conditions on Grain Quality. The effects, including at the cellular and molecular levels, of low and high air temperatures on productivity and quality of produce are described in detail. I recommend that critical temperature values be indicated. The study includes the application of innovative methods to support adaptation to climate change. The broad discussion of adaptation issues also includes Policy Implications for Climate Adaptation, where priorities for funding scientific research and mechanisms for international cooperation are outlined.
Reviewer 2 Report
Comments and Suggestions for Authors
The manuscript needs a major revision as per the suggestion attached.

Extensive English editing is required.
Reviewer 3 Report
Comments and Suggestions for Authors
Page 3, paragraph 2: summarize the paragraph.
Page 5, line195: almonds and walnuts are tree crops not horticultural production. Rectify
After page 8 there is an interruption in number pages. Please check and rectify.
Round 2
Reviewer 2 Report
Comments and Suggestions for Authors
The authors have done a good job. After addressing all the concerns I raised in my revision, I'm now able to recommend this manuscript for publication in this journal.